

# Vehicle target detection method based on improved YOLO V3 network model

Qirong Zhang[1], Zhong Han[1] and Yu Zhang[2]

[1] School of Information Science and Technology, Qiongtai Normal University, Haikou, Hainan, China
[2] College of Computer and Information Science, Southwest University, Chongqing, Chongqing, China

## ABSTRACT

For the problem of insufficient small target detection ability of the existing network model, a vehicle target detection method based on the improved YOLO V3 network model is proposed in the article. The improvement of the algorithm model can effectively improve the detection ability of small target vehicles in aerial photography. The optimization and adjustment of the anchor box and the improvement of the network residual module have improved the small target detection effect of the algorithm. Furthermore, the introduction of the rectangular prediction frame with orientation angles into the model of this article can improve the vehicle positioning efficiency of the algorithm, greatly reduce the problem of wrong detection and missed detection of vehicles in the model, and provide ideas for solving related problems. Experiments show that the accuracy rate of the improved algorithm model is 89.3%. Compared to the YOLO V3 algorithm, it is improved by 15.9%. The recall rate is improved by 16%, and the F1 value is also improved by 15.9%, which greatly increased the detection efficiency of aerial vehicles.

## INTRODUCTION

Economic development and the rapid development of motor vehicle industry technology have greatly improved people's travel conditions, but the ensuing traffic congestion and frequent accidents have also brought many adverse effects on people's lives (*Wang et al., 2023*). The rapid development of intelligent transportation systems and the relief of traffic resource scheduling pressure have become important ways to solve the above problems (*Lee & Kim, 2019*). Therefore, as the information acquisition channel of the intelligent transportation system, how to quickly and efficiently acquire vehicle information, and identify and detect vehicle targets based on video images has received extensive attention from researchers.

Combined with deep learning technology, the use of UAV video tracking and shooting can effectively expand the target detection range and make up for the lack of fixed camera capture capabilities. However, how to deal with the problem of large numbers of vehicles and small sizes has become the difficulty of UAV vehicle detection technology. *He et al. (2020)* proposed a target detection method using the Histogram of Oriented Gradient (HOG) feature, which considers factors such as context shape and scale invariance, and

Corresponding author
Qirong Zhang,
zhangqirong@mail.qtnu.edu.cn

achieves good performance results. *Redmon & Farhadi (2017)* adopted the method of deep learning, and used the YOLO network to extract the features of the image. Through improvements in normalization, anchor boxes, and the like, a Darknet backbone network with better performance has been obtained, but the performance in object recognition still needs to be improved. In 2017, *Wojke, Bewley & Paulus (2017)* fully considered the problem of multi-target tracking and recognition, and proposed the DeepSORT model. Through large-scale database learning and data association, the improved algorithm has better target detection and tracking capabilities. However, since the algorithm requires a large number of databases for training, the computational efficiency of the algorithm is low. In 2019, *Voigtlaender et al. (2019)* proposed the MOTS model. By adding the identity secondary identification module to the Mask-RCNN network, the ability to track, segment, and identify targets is greatly improved. The accuracy of vehicle recognition is over 82.5 percent. However, the algorithm's ability to adapt to UAV video needs to be strengthened.

Although some research progress has been made in the field of vehicle detection, there are still some problems such as false detection, missed detection or repeated detection of small size targets and occluded targets. In addition, in the application scenario of automatic driving, the target detection algorithm should have both real-time and accuracy, and can make timely and accurate response to any complex scene and any number of targets. The existing algorithms are difficult to be guaranteed in real-time performance and are difficult to be applied, while the first-stage algorithms such as YOLOv2 and YOLOv3 have outstanding performance in real-time performance after continuous improvement. However, there are still some difficulties in deployment. Aiming at the problem of vehicle target detection and tracking in UAV video images, this article proposes a multi-scale enhanced vehicle detection method based on the YOLO V3 network vehicle retrieval algorithm. Aiming at the problem of vehicle target detection and tracking in UAV video images, in the article, a multi-scale enhanced vehicle detection method based on the vehicle retrieval algorithm of the YOLO V3 network is presented. The features of the backbone network are enhanced to improve the model's ability to acquire and detect small target vehicles. Aiming at the problem of multi-target tracking, this article proposes a method based on location constraints. The method strengthens the correlation between the target and the trajectory, reduces the number of target ID changes, and improves the target tracking ability of the algorithm.

# YOLO V3 NETWORK MODEL

## YOLO V3 algorithm

In the aspect of target recognition, the traditional sliding window detection method has weak generalization ability and low accuracy, and the detection speed is slow. The deep learning method can effectively improve the efficiency of target detection. The RCNN target detection method proposed in 2013 can improve the speed of target detection to a certain extent through the combination of regional candidate frame and SVM classifier, but the model of this method is complex and the detection accuracy is not high. The improved Faster RCNN reduces the calculation of the model through a two-stage detection

method, but it still can not meet the actual work requirements in the real-time detection. In 2016, the YOLO model was proposed to take target detection as a regression problem, which increased the test speed of target detection to 45 frames per second, and became the main method to improve the efficiency of target detection. Considering the working environment of target detection in this article, the work of this article is to improve the YOLO model.

The YOLO V3 model is the third version of the YOLO network development, and it is an important algorithm for real-time detection of targets. Based on inheriting the advantages of the previous two versions, the algorithm has made more effective improvements in multi-target detection. The improvement in the prediction of the candidate frame reduces the confidence loss of the original network model and improves the imbalance problem of object category detection (*Tang, Wang & Kwong, 2017*). Through the increase in the number of logical classifiers and clustering categories, the prediction effect of the target category is improved, and the feature map fusion technology of multiple scales is introduced, so that the model can be applied to target detection applications of different sizes.

The YOLO V3 model divides the acquired image into several small grid units. Its model predicts a series of candidate boxes for the location information of each grid cell and calculates its confidence. Its calculation formula is as follows:

$$C_i^j = P_{ij}(Ob) * IoU_{pred}^{truth} \tag{1}$$

where $C_i^j$ is the confidence degree of the $j$-th candidate frame corresponding to the $i$-th unit. $P_{ij}(Ob)$ is a function corresponding to the detection target. $IOU_{pred}^{truth}$ is the intersection and union ratio of the prediction box and the real location box.

## Improved YOLO V3 algorithm

The YOLO V3 algorithm can meet the image detection of multi-scale targets and can meet the needs of most image detection. However, for UAV video images, the targets of video vehicle detection are all small-sized, so it is necessary to improve the small-scale detection based on the existing algorithm model.

To make the YOLO V3 algorithm more suitable for the detection of small-scale targets, the network structure adapts to the three detection scales in the original model is optimized and improved.

The feature map of scale three is fused with the second residual block of the network. Meanwhile, the scale network suitable for large targets is deleted to improve the computational efficiency of the model and the adaptability of UAV target detection (*Bruch et al., 2019*). In order to make the YOLO V3 algorithm more suitable for small-scale target detection, this article optimizes and improves the network structure adapted to the three detection scales in the original model, fuses the feature map of scale three with the second residual block of the network, and deletes the scale-one network suitable for large targets, which is to improve the computational efficiency of the model and the adaptability of the UAV target detection.

**Table 1** Allocation of anchor boxes at different scales.

| $d$ | Ranges | Anchor boxes |
|---|---|---|
| 2 | $\sqrt{h \times w} \leq 41.5$ | (10,18), (8,30), (15,35), (19,55), (25,54) |
| 3 | $41.5 \leq \sqrt{h \times w} \leq 86.4$ | (24,82), (33,60), (45,88) |
| 4 | $\sqrt{h \times w} \geq 86.4$ | (57,130) |

In the process of using the YOLO V3 network algorithm, it refers to the idea of the anchor box, and improves the training efficiency of the network model by pre-setting the size of the candidate box. The setting of the network candidate frame is mainly obtained through K-means cluster analysis, which has a good predictive ability, but also greatly reduces the convergence speed of the model.

In response to the above problems, this article uses AvgIoU as the objective function of the model to optimize the clustering analysis method. Its expression is as follows:

$$AvgIoU = \frac{\sum_{i=1}^{k} \sum_{j=1}^{n_k} IoU\left(b_i, c_j\right)}{n} \tag{2}$$

Where $b_i$ is the $i$-th real label frame of the target sample, $c_j$ represents the $j$-th cluster center of the cluster analysis algorithm, $n_k$ is the sample size corresponding to the $k$-th cluster center, $k$ is the number of clusters, and $n$ is all the number of samples, $IoU$ is the intersection ratio of the sample label box and the cluster center box.

In the optimized algorithm, the larger the number of clusters k is, the greater the detection stability of the objective function is, and the relationship between the objective function and the number of detection layers is considered comprehensively. Through the experiment summary, the number of anchor boxes selected by the improved algorithm is 9. The specific sizes are mainly (10,18), (8,30), (15,35), (19,55), (25,54), (24,82), (33,60), (45,88) and (57,130).

Considering that the sensitivity of different anchor boxes corresponding to different feature maps is very different, it needs to be placed on a feature map of the appropriate size. Thus, it needs to be allocated. According to the $IoU$ between the anchor boxes and the real frame, assuming that the height and width of the anchor boxes are $h$ and $w$ respectively, there is a relationship between the size of the anchor boxes and the number of down-sampling $d$, and the distribution results of the anchor boxes are shown in Table 1.

## Network optimization

For the initial YOLO v3 algorithm model, there is also the problem of deep networks. The main reason is that the size selection of the residual model is unreasonable, which can result in the stacking of multiple residual modules as well as affect the model calculation (*Naqvi et al., 2022*). Improving the residual module and using the combination of multiple receptive field modules can better improve the above problems. The improved residual module structure is shown in Fig. 1.

In Fig. 1, when two residual modules work on the input feature map at the same time, the convolution kernels with different receptive fields have different extraction operations on the feature map. The smaller receptive field residual module is responsible for detail

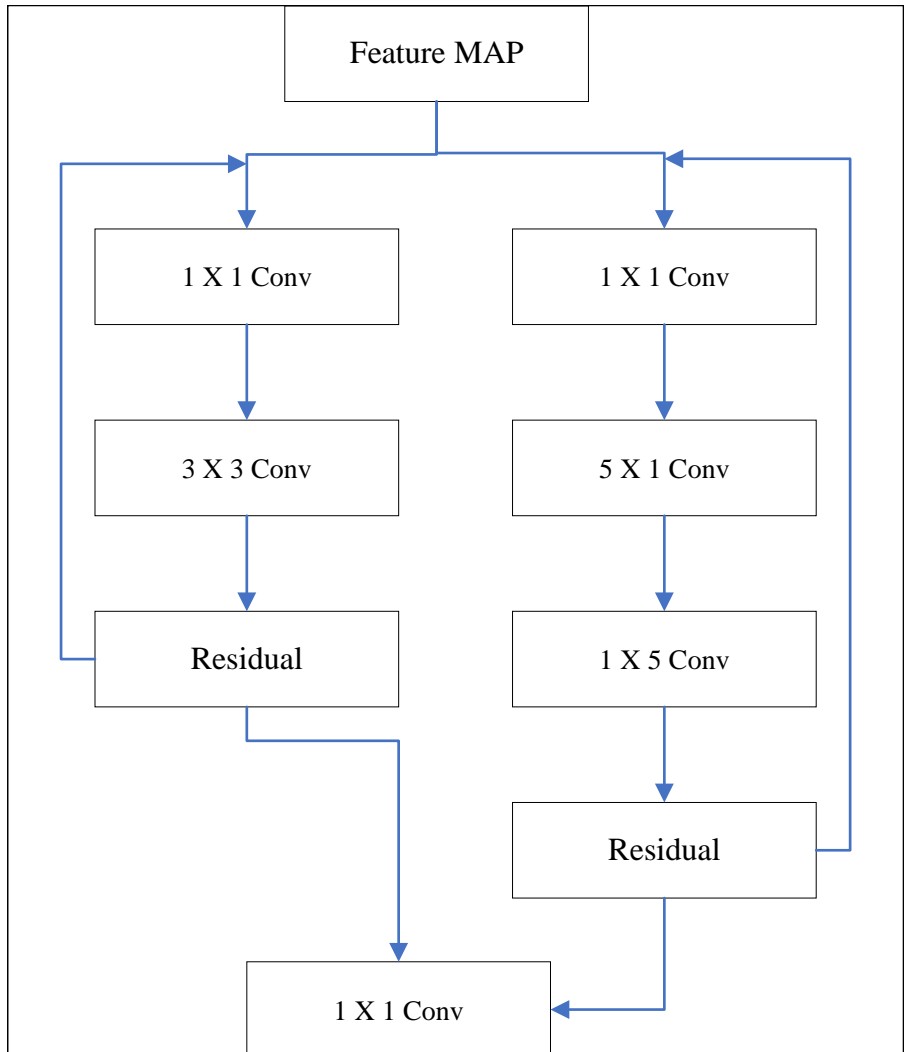

**Figure 1  Two-way residual module optimization.**

feature extraction, and the larger receptive field residual module is mainly responsible for the overall feature extraction of the target. After convolution fusion, the comprehensive features of the detection target are obtained (*Chen & Chen, 2020*). At the same time, the low-rank decomposition method is used to decompose the 5×5 convolution kernel into two serial convolution kernels of 5×1 and 1×5, which can greatly reduce the calculation amount of model analysis under the premise of ensuring the effect of feature extraction.

The influence of the number of residual modules is considered separately. For the four scales of 104×104, 52×52, 26×26 and 13×13, the number of residual modules contained in each scale is considered to be 1~4. After target testing, the mAP results were 78.5%, 88.7%, 82.1, and 72.7%, respectively. It can be seen from the results that the mAP values of the two residual models are the highest, which is better than the other three cases.

Therefore, this article chooses to use two residual modules to complete the corresponding image detection work.

## Introduction of direction angle

In the classic YOLO V3 model, the angle of the detection target is not considered. In addition, the labeling of the target is not clear and precise enough, especially when the vehicle is very close, the overlapping of the rectangular frames will lead to errors in the identification and tracking of the target. Introducing a rectangular frame with orientation angles can effectively solve the above problems (*Cho & Yoon, 2018*).

In the article, by positioning the six-dimensional vector ($b_x$, $b_y$, $b_w$, $b_h$, $\theta$, $P_r$) of the target position information, the angle of the target can be accurately reflected, where bx and by are the position coordinates of the target center, $b_w$ and $b_h$ are the width and height of the rectangular frame, $\theta$ is the direction angle of the target, and $P_r$ is the position prediction confidence. Its expression is as follows.

$$
\begin{aligned}
b_x &= \sigma(t_x) + c_x \\
b_y &= \sigma(t_y) + c_y \\
b_w &= p_w e^{t_w} \\
b_h &= p_h e^{t_w}
\end{aligned}
\tag{3}
$$

Where $\sigma$ is the Sigmoid activation function of the model, $p_w$ and $p_h$ are the width and height of the anchor box, respectively, and ($c_x$, $c_y$) are the coordinates of the upper left corner of the anchor box.

After considering the direction angle, the loss function is also improved accordingly. Therefore, the input–output loss function and the direction-angle loss function are respectively introduced and expressed by cross-entropy. The final improved loss function can be expressed as follows.

$$
\begin{aligned}
L(b_x, b_y, b_w, b_h, \theta_r) &= \lambda_{cr} \sum_{i=1}^{s^2} \sum_{j=1}^{M} W_{ij}^{ob} \left[ (x_j^g - x_i)^2 + (y_i^g - y_i)^2 \right] \\
&+ \lambda_{cr} \sum_{i=1}^{s^2} \sum_{j=1}^{M} W_{ij}^{ob} \left[ (W_{ij}^g - W_{ij})^2 + \left(h_{ij}^g - h_{ij}\right)^2 \right] \\
&+ \lambda_{P} \sum_{i=1}^{s^2} \sum_{j=1}^{M} W_{ij}^{ob} \left[ C_{ij}^g \log C_{ij} + (1 - C_{ij}^g) \log\left(1 - C_{ij}\right) \right] \\
&+ \lambda_{cl} \sum_{i=1}^{s^2} \sum_{j=1}^{M} W_{ij}^{ob} \, \text{Cross Entropy}\left(\theta_i^g, \theta_i\right)
\end{aligned}
\tag{4}
$$

In the formula, $s_2$ is the number of grid units; $M$ is the number of anchor boxes; $\lambda_{cr}$, $\lambda_{pr}$ and $\lambda_{cl}$ are the weight parameters of the corresponding losses. When the coordinates of the center point in the $i$-th anchor box are ($x_i^g$, $y_i^g$) and it contains the true center point, $W_{ij}^{ob} = 1$; otherwise, $W_{ij}^{ob} = 0$. $\theta_i^g$ is the angle between the center points, and $\theta_i$ is the angle of the $i$-th point. For multiple objects appearing in the image, each object corresponds to a bearing angle. $i$ is the number of targets in the detected image.

Where the expression formula of cross entropy is as follows.

$$\text{Cross Entropy}\left(\theta_i^g, \theta_i\right) = -\sum \theta_i^g \log \theta_i. \tag{5}$$

# EXPERIMENTAL RESULTS AND DATA ANALYSIS

## Parameter setting

The aerial photography data set VEDAI is selected as the data set for the algorithm verification of this article. VEDAI is a dataset for vehicle detection in aerial imagery as a tool to benchmark object detection algorithms in an unrestricted environment. In addition to containing very small vehicles, the database also exhibits different variability, such as multiple orientations, lighting/shadow variations, specular reflections or occlusions. The dataset contains 1,210 aerial images of $1024 \times 1024$ with a spatial resolution of 12.5 cm. There are also multiple car targets in this data set, which is suitable for the detection and verification of small targets. The data set contains nine types of targets. They include boats, cars, campers, planes, pick-up vehicles, tractors, trucks, vans and other categories. 70% of the image data in the data set are used for model training in the article, and the remaining image data are used for testing the optimized model. The stochastic gradient descent method is selected to optimize the model, and the learning rate of the algorithm is set to 0.001, and the attenuation coefficient is 0.005.

The performance evaluation of the model is measured by the recall rate, precision and F1-score, and their expressions are as follows.

$$\text{precision} = \frac{TP}{TP + FP} \tag{6}$$

$$\text{recall} = \frac{TP}{TP + FN} \tag{7}$$

$$F1 = \frac{2 \cdot \text{precision} \cdot \text{recall}}{\text{precision} + \text{recall}}. \tag{8}$$

In the formula, TP, FP, and FN are positive samples predicted as positive by the model, negative samples predicted as positive by the model, and positive samples predicted as negative by the model.

## Data analysis

### 1. Comparison of optimization algorithms

The optimized algorithm is compared with the original YOLO algorithm, and the results are shown in Table 2.

As can be seen from the results in Table 2, the algorithm in this article has a precision rate of 84.3%, which is much higher than other YOLO algorithms; and the recall rate of 83.9% is also the best result among all algorithms; the F1-score of this algorithm reaches

**Table 2  Comparison of detection accuracy of different algorithms.**

| Model | Precision | Recall | F1-score | FPS |
|---|---|---|---|---|
| YOLO V3 | 73.4% | 67.2% | 70.2% | 45 |
| YOLO V3+RESNET34 | 66.2% | 65.4% | 67.3% | 85 |
| YOLO V2 (*Couteaux et al., 2019*) | 67.5% | 61.5% | 64.4% | 48 |
| The improved YOLO V3 algorithm in the article | 84.3% | 83.9% | 84.1% | 36 |

**Table 3  Comparison of training time and detection error rate of different algorithms.**

| Model | Training time (minutes) | The anchor boxes are allocated after optimization |
|---|---|---|
| YOLO V3 | 576 | 11.3 |
| YOLO V3+RESNET34 | 564 | 13.2 |
| YOLO V2 (*Couteaux et al., 2019*) | 483 | 18.2 |
| Improved YOLO V3 algorithm in this article | 585 | 9.8 |

**Table 4  Comparison of the effect of anchor boxes on detection accuracy.**

| Model | Equal distribution of anchor boxes | Allocation after anchor boxes optimization |
|---|---|---|
| YOLO V3 | 79.4% | 74.2% |
| YOLO V3+RESNET34 | 78.5% | 76.3% |
| YOLO V2 (*Couteaux et al., 2019*) | 66.2% | 65.5% |
| Optimization algorithm after adding anchor boxes allocation | 84.3% | 82.8% |

84.1%. Since the relatively complex structure of the model, the detection speed of this algorithm is 36 f/s, which is inferior to other algorithms.

The training time and detection error rate of different algorithm models are compared, and the results are shown in Table 3.

It can be seen from the results in Table 3 that for the above models, the minimum training time of YOLO V2 model is 483 min, while that of the algorithm model in this article is 585 min. The main reason is the optimization of the model network. The optimization of the residual network structure leads to the extension of the training time. Considering the application environment of target detection, the time is still acceptable. In terms of detection error rate, due to the improvement of the detection algorithm, the detection error rate of the algorithm in this article is 9.8%, which is better than other algorithms.

Considering the influence of anchor boxes allocation on the algorithm accuracy, the comparison results are shown in Table 4.

It can be seen from the results in Table 4 that the allocation of anchor boxes can effectively improve the detection accuracy of the algorithm for small target models, increasing it from the original 79.4% to 84.3%.

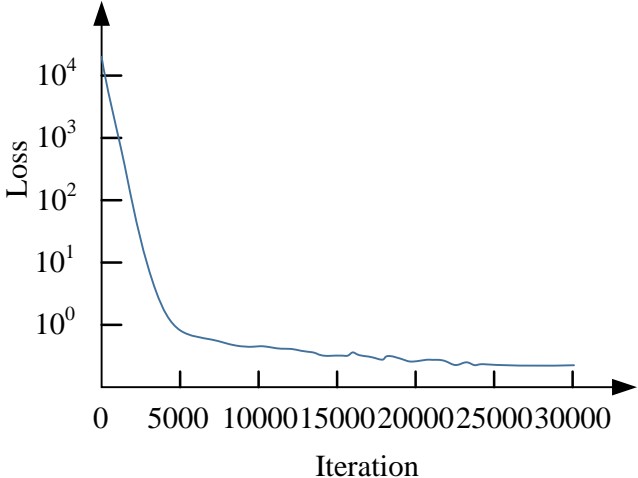

**Figure 2** Convergence curve of network loss function.

**Table 5** Comparison of detection accuracy of different indicators.

| Model | Precision | Recall | F1-score | FPS |
|---|---|---|---|---|
| YOLO V3 | 73.4% | 67.2% | 70.2% | 45 |
| YOLO V3+RESNET34 | 76.5% | 70.6% | 71.3% | 60 |
| Algorithm after optimization of two-way residual network | 88.3% | 81.4% | 84.7% | 42 |

## *2. The optimization effect of the network model*

Similarly, this article uses the previous VEDAI dataset, and the convergence curve of the loss function of its network model is shown in Fig. 2.

From the results in Fig. 2, it can be seen that when the number of iterations reaches 25,000, the parameters of the model are stable. The convergence of the classic YOLO V3 model is similar, with a relatively higher loss function. The VEDAI data set is used to conduct a comprehensive index test on the improved network, and the test results are compared with other algorithms. The results are shown in Table 5.

As can be seen from the results in Table 5, the precision rate of the improved network model is 88.3%, which is much higher than other YOLO algorithms; and the recall rate reaches 81.4%, which is still the best result among all algorithms; the F1-score of this algorithm reaches 84.7%. However, due to the complex structure of the residual module, the algorithm is slightly lacking in reasoning speed, but it is not far behind the YOLO v3 model before improvement.

The comparison of the detection results of the improved YOLO V3 network model with orientation angle detection is shown in Table 6.

From the results in Table 6, it can be seen that the detection accuracy of the algorithm model can be effectively improved through the improved model with the direction angle. The precision rate of the improved network model is 89.3%, which is much higher than the YOLO V3 algorithm, and also better than the YOLO V3 algorithm without direction

**Table 6  Comparison of detection model indicators of different algorithms.**

| Model | Precision | Recall | F1-score | FPS |
|---|---|---|---|---|
| YOLO V3 | 73.4% | 67.2% | 70.2% | 45 |
| YOLO V3-tiny (*Adarsh, Rathi & Kumar, 2020b*) | 64.8% | 59.1% | 61.9% | 68 |
| Algorithm after optimization of two-way residual network | 88.3% | 81.4% | 84.7% | 42 |
| The final optimization algorithm with improved orientation angle | 89.3% | 83.2% | 86.1% | 41 |

detection. In terms of recall rate, the recall rate of the improved algorithm model reached 83.2%, which is still the best result among all algorithms. The F1-score of its algorithm also increased from 84.7% to 86.1%. The model designed in the article improves the detection accuracy of the direction angle to the target, but due to the increased complexity of the set model, the algorithm reasoning speed is reduced. However, the overall difference is not big, and it can meet the target detection task of UAV aerial photography vehicles.

On the whole, the improved algorithm model in the article can effectively improve the detection accuracy of the algorithm model without greatly sacrificing the detection rate, and can also improve the effect of small target detection of the model, which proves the effectiveness of the improved model.

## CONCLUSION

The article improves the YOLO V3 model to make it more suitable for small target detection with aerial vehicles. Through the improvement of the detection layer and the distribution of the anchor box, the detection accuracy of the model is improved. The optimization of the network structure and the improvement of the feature extraction network have improved the efficient detection ability of the algorithm model for small target vehicles in aerial photography. Based on the improved network model, the improved model has a stronger ability to identify small target vehicles through the introduction of a rectangular prediction frame with direction angles. Through data calculation and experimental verification, the improved model can effectively locate the position information of the aerial vehicle. Meanwhile, its algorithm is sensitive enough to the direction, which can effectively improve the wrong detection and missing detection problems of the original model. The accuracy of the improved model detection is 89.3%, which is better than that of the YOLO V3 algorithm without direction detection, and is improved by 15.9% compared with the YOLO V3 algorithm. In terms of recall rate, the improved algorithm improves the recall rate by 16% and the F1 value by 15.9%, which is the best result of all algorithms.

### Funding

This work is supported by Topological vector space identification and its application in multi-network vehicle intelligent diagnosis in Hainan Province of China (Grant No.

722RC740). The funders had no role in study design, data collection and analysis, decision to publish, or preparation of the manuscript.

## Grant Disclosures

The following grant information was disclosed by the authors:

Topological vector space identification and its application in multi-network vehicle intelligent diagnosis in Hainan Province of China: 722RC740.

## Competing Interests

The authors declare there are no competing interests.

## Author Contributions

- Qirong Zhang conceived and designed the experiments, analyzed the data, authored or reviewed drafts of the article, and approved the final draft.
- Zhong Han conceived and designed the experiments, performed the experiments, performed the computation work, prepared figures and/or tables, authored or reviewed drafts of the article, and approved the final draft.
- Yu Zhang performed the experiments, analyzed the data, performed the computation work, prepared figures and/or tables, and approved the final draft.

## Data Availability

The algorithm model is available in the Supplemental File. The data is available at Vehicle Detection in Aerial Imagery (VEDAI): a benchmark: https://downloads.greyc.fr/vedai/.

## Supplemental Information

Supplemental information for this article can be found online at http://dx.doi.org/10.7717/peerj-cs.1673#supplemental-information.

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
