# Peer review of "Vehicle target detection method based on improved YOLO V3 network model"

_PeerJ Computer Science, doi:10.7717/peerj-cs.1673_

## Round 0.1 · original submission · Major Revisions

According to the comments raised by the reviewers and my reading of the manuscript, I can make the major decision. The authors need to focus on the experiment parameters, format issues and some description in formulas, concept and methods, etc.

Reviewer 1 ·

Basic reporting

The work present a modified yolo model for vehicle detection. The presentation of the work is not done properly and could be better. The author need to answer few queries:
1. Why yolo model only why the work is not compared with any other trained model
2. the work which are refered and cited in the work are too old. Author need to compare the work with some recent work and more recent work in review section .
3. An impropriate description of the author contribution is give. author need to highlight what are their contribution and at which position in yolo the changes are done.
4. Objective of this work need to be highlighted and how will this be usefull in real world. Simple improving one parameter like accuracy/ precision is not acceptable.
5. Need to check whether the model is not going under overfitting or underfitting?

Experimental design

The finding are acceptable but need to compare the work with more recent and most cited work in this field. Auhtor has given no description of the database features and any other existing work on this database. What is the importance of this database? is is a standard database.

Validity of the findings

Need to give more results to support their conclussion.

Cite this review as

Reviewer 2 ·

Basic reporting

In order to strengthen the detection of small vehicle targets, this paper improves the YOLO V3 network model, introduces the orientation angle, and improves the training efficiency of the network model by presetting the size of the candidate frame. It has certain theoretical significance. But there are following problems:
1 Figure 2 shows the convergence curve of the network loss function of the improved model. Under the same conditions, whether the convergence of other models is fast or slow, it is recommended to add the convergence curve of the comparison model in Figure 2.
2 In formulas 4 and 5, what is the range of the letter i, and the letter g should indicate the corresponding meaning.
3 The aerial photography data set VEDAI contains 9 types of targets, which 9 types?
4 Individual sentences need to be revised, and it is recommended to proofread the grammar of the full text.
5 Some of the cited documents need to be summarized in one step, and the problems existing in them should be pointed out, not just an overview of the content of the documents.

Experimental design

no comment

Validity of the findings

no comment

Cite this review as

Reviewer 3 ·

Basic reporting

This paper proposes an improved YOLO V3 network model for vehicle target detection, which has certain research value, but there are some errors, and it is recommended to accept after major revision.
1. In the introduction, you should increase the organizational structure of the paper, and at the same time, also increase the part of related work, and put some references of the introduction into the related work. In addition, it is necessary to summarize the referenced documents, not a simple statement.
2.There are some grammatical errors that require further refinement of the language.
3. In the positioning of the six-dimensional vector (bx, by, bw, bh, θ, Pr) of the target position information, θ is the direction angle of the target, where the direction angle is fixed? Does it cover most targets well?
4. Is the statistical data analysis of this paper performed in accordance with the technical standards required for publication?
5. What is the sample size required for the experiment? Are you using a public dataset or a dataset you collected yourself?
6. The format of the references is not uniform, and individual documents lack information such as volumes and issues. At the same time, references from the past two years have been added, and some documents before 2015 have been removed.
7. In the experimental results section, the data value should be used to illustrate the superiority of the method, such as how much it has been improved or reduced.
8. The formula numbers are not standardized, and the formula numbers are aligned to the right.

Experimental design

no comment

Validity of the findings

no comment

Cite this review as

---

## Round 0.2 · Major Revisions

Dear authors,

Please revise your paper again. One of the reviewers isn't satisfied with the revised version and replies.

Reviewer 1 ·

Basic reporting

the work is not satisfactory which lacks in identifying the clear objectives and methodology. the work is simply a performance study of different YOLO models on vehicle target detection data.

Experimental design

The design is not truly impresive as existing yolo model has been tested with new data. But i would be intrested if the YOLO would be retrained on this labelled data. Then study the performance.

Validity of the findings

the findings are not very clear as the results only discusses the comparison of different yolo model that exisit. but that accuracy can be over fitting under fitting or many other reason the study on that need to be performed with existing models like RESNET using various other parameters

Cite this review as

Reviewer 2 ·

Basic reporting

Authors addressed all the comments provided by the reviewers. The quality of the paper is improved a lot, contribution and novelty is clear.It can be accepted in this form.

Experimental design

no comment

Validity of the findings

no comment

Cite this review as

Reviewer 3 ·

Basic reporting

no comment

Experimental design

no comment

Validity of the findings

no comment

Cite this review as

---

## Round 0.3 · accepted · Accept

Dear authors,

It is a pleasure to inform you that your revised version of the paper is satisfactory and we can accept it.